# Mechanical Properties of Flax Tape-Reinforced Thermoset Composites

**DOI:** 10.3390/ma13235485

**Published:** 2020-12-01

**Authors:** Forkan Sarkar, Mahmudul Akonda, Darshil U. Shah

**Affiliations:** 1Department of Textile Engineering, Dhaka University of Engineering & Technology, Gazipur 1700, Bangladesh; forkan@duet.ac.bd; 2School of Materials, The University of Manchester, Manchester M13 9PL, UK; 3Department of Architecture, University of Cambridge, Cambridge CB2 1PX, UK; dus20@cam.ac.uk

**Keywords:** natural fibers, IFSS, microstructural analysis, unidirectional tape

## Abstract

Three thermoset resin systems—bio-epoxy, epoxy, and polyester-with 30 v% flax fiber reinforcement have been studied to identify the optimal fiber–resin combination in a typical composite structure. Tensile, interface and interlaminar shear strength together with flexural and impact damage tolerance were compared in this study. The results revealed that mechanical and interfacial properties were not significantly affected by the different resin systems. Microscopy studies reveal that epoxy laminates predominantly fail by fibre linear breakage, polyester laminates by fiber pull-out, and bio-epoxy laminates by a combination of the two. The higher failure strains and pull-out mechanism may explain the better impact damage tolerance of polyester composites. Flow experiments were also conducted, revealing faster impregnation and lower void content with polyester resin, followed by bio-epoxy, due to their lower viscosities. Overall, bio-epoxy resin demonstrates comparable performance to epoxy and polyester resins for use in (semi-)structural bio-composites.

## 1. Introduction

The composites industry has drawn significant attention to utilizing the benefits of lightweight, low-cost natural fibers such as flax, jute, kenaf, hemp and sisal to replace glass fiber in specific stiffness-limited design applications [1]. In addition, natural fibers have excellent vibration and noise insulation properties. Consequently, in the past two decades, European auto-manufacturers have used natural fiber composites with thermoplastics and thermoset matrices for interior parts such as door and instrumental panels, seat backs, headliners, package trays, and load floors [2]. In those applications, carbon and glass fibers are the main choice for the composite manufacturer. Though these synthetic fibers have a significant environmental impact as they are made from petroleum-based materials. In addition, manufacturing costs, together with the health hazards have drawn great attention to composite manufactures [3]. Jute, flax, hemp, sisal, kenaf and ramie are mainly used as an alternate of glass fiber in stiffness driven applications [1]. Amongst all other bast fibers, flax fiber, is more likely to be used in the automotive industry as it grows abundantly in the European continent, particularly France. Besides, the thermoplastic matrix is very popular in automotive composite parts manufacturing due to the low processing time and temperature. On the other hand, a simple vacuum resin infusion process in making complex automotive parts has widened the opportunity to use thermoset matrices in many applications. Moreover, thermoset matrices have a range of benefits over thermoplastic matrices such as: higher mechanical properties, lower viscosity that allows low-temperature requirement in processing, better interfacial shear strength due to the better polarity between the functional group of thermoset matrices and natural fibers [4,5]. In cases of environmental impact, both of the matrices need almost similar energy in recycling [6]. Several researchers have reported on the mechanical performance of different resins systems for bio-composites. Rassman and Paskaramoorthy [7] investigated the effect of three resin systems (epoxy, vinyl-ester and polyester) on kenaf fiber and found no significant effect on the tensile and flexural properties of the composites. Salman et al. [8] made similar observations and found better adhesion of epoxy matrix with kenaf woven fiber. Shah et al. [4]. also found better tensile properties with epoxy matrix than the polyester resins while reinforced with jute fiber. Scarponi et al. [9] recently evaluated epoxy and bio-epoxy on the impact damage response of hemp fiber-reinforced composites and found improved toughness and damage tolerance for bio-epoxy based composites compare to traditional epoxy composites due to better adhesion bio epoxy with hemp fiber. However, the complete evaluation for the effect of matrix types are still limited in the literature. Inhomogeneous mode of natural fiber needs more study to make select the best resin system for a particular fiber to a required application.

Besides the resin system, it is also necessary to select the best fiber architecture to tailor the performance of natural fiber composites. To utilize maximum mechanical properties of fiber in composites, all the fibers have to be placed parallel to the loading direction [3]. Natural fibers are not convenient as synthetic fibers to be arranged in parallel. Numerous studies have been conducted recently to maximize the orientation of fiber reinforcement intermediate products, which includes developing wrap spinning, automated fiber placement, dynamic sheet forming, and oriented flax tape [10,11,12,13]. Previous studies proved that newly developed highly aligned flax tape offer maximum mechanical properties compared to traditional textile-based natural fiber composites. Therefore, innovative, high-performance and unidirectional flax tape have been selected in his study. On the other hand, interfacial shear strength (IFSS) is considered to be responsible for transferring stress to the load-bearing fibers [1,4]. Most importantly the interfacial bond strength indicates the energy-absorbing characteristics of the interface [14]. The microbond test [1] is commonly used to characterize the IFSS of natural fiber micro-composites. However, microbond study against different resin systems is less reported in the literature. Therefore, the test employed in this study to get the role of interface between the flax fiber and different thermoset matrices. The main objectives of this study were to identify the best of three different thermosetting resin systems—bio-epoxy, epoxy and polyester—considering matrix/fiber compatibility, stiffness, strength, and low-velocity impact performance of flax tape reinforced composites. As part of the investigation of mechanical properties of the composites: tensile, flexural, IFSS, interlaminar shear strength (ILSS) and drop-weight impact tests were conducted.

## 2. Materials and Methods

### 2.1. Martials

#### 2.1.1. Flax Fibre Tapes

Unidirectional flax tape (150 g/m^2^) was sourced from Tilsatec Ltd, Flanshaw Lane, UK. All the flax fibers used in this process have a unidirectional arrangement without any twist. A small percentage of poly-lactic acid polymer binder (3 w%) was used to hold the flax in tape format. The fiber assembly was passed through a heating and pressing unit during the manufacturing of the tapes (see Figure 1). The flax fibers were otherwise untreated and used in their as-received form.

#### 2.1.2. Bio-Epoxy, Epoxy and Polyester Matrices

Three standard thermoset resins were used in this study: SuperSap CLV Clear Epoxy Bio Resin (resin: hardener-100:40), EL2 Epoxy Laminating Resin (resin: hardener-100:30), and low viscosity pre-accelerated isophthalic IP2 polyester resin with 2% MEKP catalyst. The physical, mechanical and flow properties of the different resins are given in Table 1.

### 2.2. Methods

#### 2.2.1. Composite Manufacturing

Twelve layers of flax tape preform were cut to size (170 mm × 350 mm) and then cross-laminated in 0/90 fashion. The sequenced laminates were placed on the mold surface, sandwiched between layers of peel ply. A net mesh infusion medium was placed on top to ensure uniform flow distribution. A nylon bag was used to seal the reinforcement under full vacuum, with the inlet pipe clamped. The inlet pipe was then placed into the de-gassed resin bath to resume the vacuum reinfusion process. According to the manufacturer’s instruction, the infused panel was left with a sealed bag for 24 h at room temperature in order to fully cure the composites panel. Figure 2 illustrates the set-up. Multiple samples that have a thickness of approximately ~3 mm were produced for repeatability. Details of the composites produced in this study are given in Table 2.

#### 2.2.2. Infusion Process Study

Permeability is one of the important criteria in any composite molding process in order to predict the flow of resin. An experimental study was conducted to track the flow front against time in the vacuum assisted resin infusion process (see Figure 2). A high definition digital camera was used. The flow front (*x*) and filling time (*t*) were recorded, according to the recommendations of Subbiah et al [15], to monitor the infusion process and calculate the unsaturated permeability *K_unsat_* of the flax tape. Unsaturated or transient permeability *K_unsat_* refers to the permeability of fabric while it is being freshly impregnated (i.e., full saturation has not occurred). *K_unsat_* can be calculated by rearranging Darcy’s law into Equation (1) [16], where *x* is the position of the flow front at time *t*, *ϕ* is the fabric porosity, Δ*P* pressure gradient applied, and μ is the resin viscosity (in Table 1). Fabric porosity *ϕ* is directly related to the fiber volume fraction (1-*vf*), and in our case *ϕ* = 0.70–0.73 (Table 2), and based on the full vacuum applied during infusion, Δ*P* = 105 Pa. Plotting the square of the flow front position *x*^2^ against fill time t would produce a straight line graph [16], and the slope of the graph *x*^2^/*t* can be used as an input in Equation (1) to determine *K_unsat_*.
(1)$$K_{unsat}=\frac{x^{2}}{t}\frac{\Phi \mu }{2\Delta P},\;\mathrm{where}\;\Phi \equiv 1-v_{f}.$$

#### 2.2.3. Volume Fraction and Density Calculation

The volume fraction of the composites was measured by taking into account the weight of the dry fiber W_f_ and the manufactured composites W_c_. Archimedes principle [17] was employed to measure the density of the composite ρ_c_ with the specimen chamber at 20 ± 1 °C. Detail of the study can be found in our previously reported work [18].

#### 2.2.4. Optical and Scanning Electron Microscopy

Optical microscopy was used to qualitatively measure the image of fiber packing arrangement and porosity of the composites. For this, three sections from each of the composites were cast using epoxy resin (resin to hardener ratio 100:10 by mass) and cured for 48 h. The samples were polished sequentially using 240, 400, 600, 800, 1200 grit paper, followed by diamond grit paper of 6 µ and 1 µ. Finally, these polished samples were viewed under a microscope at ×250 magnification. Images were processed using ImageJ software. In addition, fractured composites were sputter-coated with gold in an Edward Sputter Coater and then observed under a Zeiss EVO50 Scanning Electron Microscope (SEM).

#### 2.2.5. Micro-Bond Testing

Single fibres were collected from the technical flax fiber and then mounted on a paper frame with the help of super glue on it. Sample for a micro-bond test was prepared by following the procedure described in previous works [1,19]. We used Zwick/Roell, UK tensile testing machine to perform this test where a 20 N load cell was used and cross-head speed was maintained at 0.25 mm/min. Debonding fore, *F* was recorded for each sample during the test and following Equation (2) was used to calculate the IFSS.
(2)$$\tau_{IFSS}=\;\frac{F}{\pi Dl},$$
where *τ_IFSS_* is the measured interfacial shear strength and *D* is the diameter of the fiber and *l* droplet length measured before performing the test.

#### 2.2.6. Tensile Testing

Mechanical testing was conducted using ASTM D3039 standard to measure the tensile properties of the composites. Specimen dimensions were 250 mm long and 25 mm wide for each category of the composites. An Instron 5982 model machine (UK) was loaded with 100 kN load cell for performing the test. We used a constant crosshead speed at 2 mm/min for doing the test. Young’s moduli of the composites were calculated from the slope of stress–strain curve from 0.1 to 0.3%.

#### 2.2.7. Flexural Test and Short Beam Shear Test

We performed three-point bending test in an Instron 5969 (UK) machine by following the standard ASTM D-790. A nominal sample size of 127 mm length and 12.7 mm width support span of 96.3 mm were used for this test. The machine was loaded with a 10 kN load cell and the speed of the machine was kept 5.1 mm/min. We also performed ILSS with the same machine and load cell by following the standard ASTM D2344. Samples were prepared by determining the span to thickness ratio of 4 and length to thickness ratio of 6. The crosshead speed was 1 mm/min with a 2 kN load cell.

#### 2.2.8. Low-Velocity Impact Test

A low-velocity impact test was performed according to ASTM D7136 at two impact energy levels (5 J and 10 J). Lower energy levels were selected as the preform that we used was unidirectional and there was no crimp and twist in the preform which could absorb more energy. An Instron 9350 drop-weight impact machine was used to measure the impact of resistance against penetration and the absorbed energy. Details of the test can be found in previously published work on hemp fiber composites [9,20]. We used a Midas-NDT ultrasonic C-scan to measure the damaged area after the impact test. We then used a digital meter (depth gauge) to measure the dent depth of the tested composites.

## 3. Results and Discussion

### 3.1. Composite Fill Time and Flax Tape Permeability

We observed that the infusion time across the cross-laminated flax tape differed for the various resin systems. The resin filling time was longest for epoxy resin (see red arrow mark in Figure 3b) and the shortest filling time was observed for polyester resin (see Figure 3c). The reason for this observation is the lowest viscosity of polyester resin (160 mPa∙s) allowing rapid impregnation of fiber tows. On the other hand, epoxy resin has almost an order of magnitude higher viscosity (1200 mPa∙s) leading to slow impregnation. In contrast, the bio-epoxy resin-with viscosity of 800 mPa∙s-has a 50% lower filling time than the epoxy resin. The flow front position for the different resins at two minutes is visible in Figure 3a–c (marked with a red line). A dual scale flow is noticed. During the infusion process, the resin first passes longitudinally through the inter-fiber channels without impregnating the fibers. Once the flow front has passed, the resin in the entrapped channels starts to impregnate the fibers in the transverse directions.

Using Equation (1) with the relevant inputs for fabric porosity (0.70–0.73), applied pressure gradient (105 Pa), and the slope of the graph *x*^2^/*t* (Figure 4) for the resins with their respective viscosities (Table 1), we were able to calculate the mean unsaturated permeability of the flax tapes *K_unsat_* to be 2.3 × 10 − 10 ± 1.0 × 10 − 10 m^2^. This is comparable to values of *K_unsat_* for other plant fiber reinforcement textiles at fabric porosity of ca 0.70–0.73 [16].

### 3.2. Density and Fibre Volume Fraction

Matrices that we used in this study have a comparable density (1.1–1.2 g/cm^3^, Table 1) therefore, we find that the resulting composites also have similar densities (in the range of 1.21 to 1.25 g/cm^3^, Table 2). The fabricated composites have fiber volume fractions ranging from 27 to 30% (Table 2). This relatively low fiber volume fraction is a constant issue for natural fiber composites as the mechanical performances of composites generally improves with increasing fiber volume fraction, and fiber volume fractions typically exceeding 50% are desirable for (semi-)structural applications. Natural fiber reinforcement products are composed of discontinuous fibers with non-uniform and non-circular cross-sections, which are typically arranged in bundles (Figure 5a–c). The selected composite manufacturing technique has a governing role in the achievable fiber volume fractions through compaction and consolidation. In a compression or hot press molding process, high fiber volume fractions can be achieved by compressing the preforms to a desirable thickness at high pressure (several bars). However, in this work, vacuum infusion is employed because of its cost-effectiveness, particularly in manufacturing composites with complex shapes, and the ability to reduce porosity content. To obtain higher volume fractions, vacuum-assisted resin transfer molding or pre-pregging methods may be considered.

### 3.3. Reinforcement Packing and Composite Porosity

Figure 5a–c shows the cross-sectional images of composites manufactured with three resin systems. The flax fiber bundles distribute homogenously in the three matrices. The images also depict the irregular cross-section and diameters of emanatory flax fibers and their technical fiber bundles. More fiber agglomeration and resin-rich areas are visible in the B-flax and E-flax composites than in the P-flax composites. This might be related to the viscosity of the resin and the interaction between the functional group of resin and harder. In this regard, Epoxy resin has a higher viscosity and better interfacial shear strength as discussed earlier allowed the fiber to be agglomerated more whereas polyester resin has very low viscosity and flashed out during impregnation (see Figure 3c) together with the lower interfacial interaction made comparatively lower fiber packing in the composites.

The void content of flax tape composites ranges from ~1–2%, calculated using Equation (1), with polyester composites showing the lowest void content of 1%. As Madsen et al. [21] observed that porosity of natural fiber composites has logarithmic relation with a viscosity of matrices.

In this regard, the polyester resin has the lowest viscosity of 160 mPa∙s (Table 1), which can explain its low void content. Generally, porosity less than 1% mandator for high-performance composites but low-performance composites having porosity ranged from 4–5% can be used in automotive and marine applications [3]. Similar results with porosity in other studies based on natural fiber composites are reported in the literature [22]. Though, it is important to identify the category of porosity present in the composites in order to take further corrective measurements during the manufacturing of composites. In this regard, Madsen et al. [23] described the four main types of porosity that exist in natural fiber composites: (I) fiber-related porosity, (II) interface porosity, (III) impregnation porosity, and (IV) matrix porosity. In this study, we find luminal porosity or fiber porosity has a contribution in all of the flax composites as this is inherent in flax fibers (Figure 5a–c). This luminal porosity has a significant contribution to the crack initiation of unidirectional natural fiber composites loaded in the transverse direction [24]. Cross-sectional images of bio-epoxy and epoxy flax composites (B-flax, E-flax) also show substantial interfacial and impregnation porosity. In this case, the elementary fibers are assembled more compactly in the technical fiber that causes low permeability of resin within the fiber bundles. Matrix porosity was trivial, probably due to the vacuum infusion manufacturing process utilized.

### 3.4. Interfacial Properties

Interfacial properties of natural fiber composites are considered to be one of the dominating characterization of composites which can show dominancy in other primary mechanical properties such as flexural and impact properties. We used the microbond test method to evaluate the IFSS of flax with the three resin systems. Figure 6a presents a typical interfacial shear force curves of the various resin microdroplets debonding from flax fibers. Figure 6b presents the IFSS recorded for the three resin systems, ranging between 4.1 MPa (polyester) and 5.4 MPa (epoxy) (see Figure 6b). A significant variation in the IFSS value is observed, leading to results being not statistically significant (*p* > 0.05, two-tailed *t*-test). The large variation may be partly related to the variation in fiber dimensions, as well as inherent flaws and defects of natural fibers [25] (vis. lamellar structure, kink bands). In general, we observe higher IFSS for epoxy resins (bio-epoxy and synthetic epoxy), probably due to a good affinity between flax fiber surface hydroxyl groups and amine and epoxide functionalities in epoxy [26]. On the other hand, polyester composites exhibit comparatively weaker bond strength. It is also notable that for epoxy, dynamic friction is not visible in the force-extension curve, though for bio-epoxy and polyester resin frictional shear force is visible (see Figure 6a). When interface bonding is relatively weak (B-flax, P-flax) debonding occurs at low interfacial force and frictional sliding occurs more radially compare to a strong interface (E-flax). This can lead to the onset of the damage mechanism of the composites which is discussed in the next section of this paper. This observation is particularly interesting noting that epoxy composites were more prone to impregnation and interfacial porosity than polyester composites (Figure 5a–c), although perhaps the void content is still low (1–2%) to make it a predominant factor.

### 3.5. Interlaminar Shear Properties

Interlaminar shear strength (ILSS) is determined to know about the strength of composites upon transverse loading on composite laminates. For interlaminar shear, the mode of failure depends on the span to depth ratio. It is well known that when subjected to three-point bending, short, stocky beams usually fail by shear along/around the neutral axis, and long, slender beams by tension or compression at the surfaces [27]. In the former, there is a combination of shear failure modes like fiber rupture, micro-buckling and interlaminar shear cracking [28]. In our experiments, no delamination of plies occurred. Rather, failures of specimens occurred due to bending, simultaneous breaking of fibers and partial pull-out of fibers [27]. For this reason, it is sometimes difficult to interpret the results of short-beam shear tests. Nevertheless, trends in ILSS are comparable to those observed for IFSS, presumably due to better impregnation and interfacial bonding. Epoxy flax composites (E-flax) exhibit ILSS of 21.6 MPa, 13% and 21% more than the bio-epoxy and polyester composites (see Figure 6c), though the differences are not statistically significant (*p* > 0.05). Previous studies [27,29] suggest that ILSS is dominated by matrix properties, as the matrix often fails first in such a test. Epoxy resin has relatively higher tensile properties (see Table 3) and forms a stronger interface with flax fibers (IFSS—Figure 6b). Our observations confirm that ILSS is indeed primarily dependent on matrix properties rather than fiber properties.

### 3.6. Tensile Mechanical Properties

Figure 7a,b and Table 3 present the measured tensile properties of the various flax tape thermoset composites. Though single fiber tensile properties of flax fiber have a large scattering effect due to irregularity in the fiber fineness at the composite scale we did not find any such significant variations in the tensile properties. The coefficients of variation of the properties were less than 10%. In tensile loading conditions, the resin plays a significant role in transferring loads across the fibers in the composites. Besides the tensile properties resins also plays a significant role in the adherence of reinforcing materials [1]. Though the actual mechanical properties of composites materials mainly depend on the stiffness and strength of the reinforcing materials [3]. One study based on resin systems was conducted by Shah et al. [5] to see the efficacy of epoxy and polyester resin systems on jute composites’ mechanical properties. Their results showed that epoxy composites showed better tensile strength than the jute polyester composites. Joffe et al. [30] found almost similar tensile properties of flax fiber composites reinforced with three different resin systems (polyester, epoxy and vinyl ester) despite having a great difference in the tensile properties of the resin itself. They recommended the need for further investigation on the fiber/matrix interface and the load transfer mechanisms for predicting fracture toughness.

While Madsen et al. [21] found noticeable differences between the tensile properties of hemp yarn thermoplastic composites based on different matrices (PE, PP, PET). However, they reported that the fiber/matrix interface has a minimal effect on the tensile properties of the composites because in the unidirectional arrangement of the composites the maximum load is transferred through the length of the fiber, and fiber properties dominate.

Just as there are some differences in tensile properties of the matrix (Table 1), differences in tensile properties of the various flax tape composites are observed (Figure 7a,b and Table 3). However, these differences are not statistically significant (*p* > 0.05). The composites have tensile stiffness in the range of 6.5–7.1 GPa and tensile strength in the range of 107–116 MPa. Indeed, in terms of tensile stiffness and strength, bio-epoxy composites have comparable, if not slightly higher, properties than polyester and epoxy composites. In this comparison, we do note the slightly lower fiber content in epoxy composites, though we do not expect this to significantly alter the general conclusion. We do observe a statistically significant (*p* < 0.001) difference in the tensile failure strains of epoxy and polyester composites, but not bio-epoxy composites. Intriguingly, the failure strain of polyester composites (2.02%, Table 3) is higher than that of epoxy composites (1.85%), despite the failure strain of the polyester resin (2.5%, see Table 1) being much smaller than that of epoxy resin (6%). This may in part be explained by the differences in failure mechanisms, described below, as well as the fact the composite failure strain is governed principally by flax fiber failure strain (around 1.8–2%). All the tensile fracture specimens show a smooth brittle fracture, however, we also observed both the fiber pull-out and matrix debonding which is perpendicular to the fiber directions.

Figure 8 shows the broken surfaces of tensile tested flax composites made from bio-epoxy, epoxy and polyester matrix. We observed more fiber pull-out in polyester laminates, more catastrophic fiber fracture in epoxy laminates, and bio-epoxy laminates exhibit both of the fiber pull-out and fiber breakage. We noticed the failure strain of polyester composites was higher than the others, as fiber debonding and pull-out was responsible for enabling larger deformations. Pull-out of fiber is an indication of a weak interface while fiber breakage indicates good stress transfer between matrix and fibers [31]. Hence, this also confirms the general trend that flax/epoxy have better interfacial bonding than flax/polyester, with flax/bio-epoxy having intermediate performance (Figure 6b,c). interfacial bonding with flax fiber.

### 3.7. Flexural Mechanical Properties

Flexural properties are influenced by the combination of a material’s tensile, compressive and shear properties. Failure in flexure occurs when any one of the three primary stress reaches its limit value. Flexural properties obtained in the three-point bending test from various flax tape composites are listed in Table 4 and Figure 7c,d. While flexural modulus is comparable at 9.4–12.1 GPa for all the flax thermoset composites (i.e., no statistically significant difference, *p* > 0.05), flexural strength and failure strain are significantly different in epoxy and polyester composites. Epoxy composites show the highest strength of 144.5 MPa compared to 115.6 MPa for polyester composites.

In contrast, polyester composites had the highest failure strain of 3.38%, with epoxy composites at 2.81%. It is obvious that the specimen under flexural loading failed with the combination of interlaminar fracture and or compressive failure can be seen in (Figure 9a–c) [32]. Generally, compressive failure mode occurs in the resin reach area, the point at where warp-weft fibers intersect in the upper surface layer [32]. In contrast, interlaminar shear fractures manifest in the lengthways towards the center of the span length. In bio-epoxy and epoxy flax composites, the interlaminar shear fracture was more common and appeared along the full length of the tested specimens, whereas for polyester composites, such shear fractures occurred towards the ends of the specimens. Based on failure theory [33,34], the results in Figure 9a,b shows that transverse cracking and delamination damage occurred more readily in the laminate made from bio-epoxy and epoxy composites than the polyester composites. This might be related to the improved interfacial strength of bio-epoxy and epoxy resin with flax fibers due to its better chemical affinity and good wettability towards epoxy resins. On the other hand, polyester resin shows relatively poor IFSS with flax fibers as discussed in the previous section. Notably, the trend in flexural properties is the same as the trend in interfacial and interlaminar shear properties can confirm the relation that the better the interfacial shear strength, the higher the flexural properties of composites.

### 3.8. Impact of Mechanical Properties

The falling weight impact (IFW) test is one of the test methods used to assess the impact properties (fast fracture or rapid crack propagation) of composites. There are a limited number of studies in the literature assessing the damage tolerance properties of natural fiber composites. However, this test is one of the essential ways of assessing the suitability of composites materials for (semi-)structural application. We conduct tests at two impact energies (5 J and 10 J) to investigate the damage process in flax tape thermoset composites. Load-deflection curves obtained from the impact test can provide some meaningful information to evaluate the damage tolerance of composite structures [35]. Figure 10 shows the load-displacement curves at the penetrating impact level of 5 J and 10 J, respectively, and Table 5 lists the measured impact properties (peak force, displacement, absorbed energy). At an impact level of 5 J, epoxy and bio-epoxy composites show several peaks in the force-displacement curves before failing in a brittle manner (Figure 10a). In contrast, polyester flax composites show the highest peak load, but do not exhibit any peaks suggesting progressive failure. In general, absorbed energy for all the composites falls in a narrow range between 4.2–4.4 J, and maximum deflection between 5.3–5.9 mm (Table 5), and any differences are not statistically significant (*p* > 0.05). The only statistically significant difference (*p* < 0.05) is in the peak force for bio-epoxy composites, which is found to be much lower than that of epoxy and polyester composites.

At 10 J impact, energy epoxy and polyester composites once again outperform the bio-epoxy composites in terms of peak loads (Table 5, Figure 10b). In addition, the bio-epoxy composites exhibit the highest maximum deflection (10.1 mm), significantly higher than that of epoxy composites (at 4.2 mm). Indeed, substantial internal damage is observed for bio-epoxy composites, confirmed by observing the peak load which is extended at peak (see Figure 10b). It was observed that, when we increased impact energy we obtained a broad area under the curves which indicates the ability of the laminates to absorbed more energy and induced damages [35,36].

Moreover, Sarasini et al. [35] explained that the difference in the load-deflection curve indicates different failure modes (tensile and shear properties). Besides, they pointed out some of the points in the load-defection curve such as, the initial load drop is responsible for the initiation of matrix crack and delamination, second load drop at peak confirms the matrix cracking, fiber breakage, delamination and the damaged area [35,37]. In agreement with this statement, particularly, in the first damage point of view, epoxy composites enhance the peak load and reduce deformation. Bio-epoxy composites show the highest deflection in tests at both energy levels (Table 5). No penetration is visible for the samples tested under 5 J and 10 J impact energy. It is noticeable that after the impact test, the rear side of the specimen showed the greater intensity of damage than the impacted face. All the tested samples exhibit cross-shaped damage (Figure 11a–f). We further inspected the samples by CT-scanning (Figure 11a–f) to confirm the physical damage modes of the composite laminate. Comparing all composites, polyester composites show the smallest damaged area (6 and 7.8 cm^2^ at 5 J and 10 J, respectively). For the bio-epoxy composites, the damaged area under both energy levels is much higher (15 and 25.5 cm^2^). No inter-ply delamination is visible for any of the tested samples which may occur due to non-woven architecture that generates fiber flaws between 00 and 900 directions. With regards to energy absorption, polyester composites offer the best performance (see Figure 10). However, epoxy composites exhibit comparable (statistically significantly) energy absorption in terms of absorbed impact energy, though global ductility is higher for the polyester composites. In contrast, bio-epoxy composites show a more pronounced localization of impact damage (see Figure 11a,b). When we plot dent depth (depth of damaged area in ~3 mm thick laminate) against applied impact energy (Figure 12) we observe a positive correlation between the applied impact energy and dent depth. Dent depth is significantly more prominent for polyester composites, whereas epoxy and bio-epoxy composites have relatively low and comparable dent depths, making these more suitable for applications where damage depth needs to be limited/minimized.

## 4. Conclusions

A comparative study of the manufacturing, mechanical and interfacial properties of composites made of flax tape and three different thermosetting resins, namely bio-epoxy, epoxy and polyester, was investigated. All composites were produced with low (<25) void content, with polyester composites, owing to the low viscosity of polyester, exhibiting the lowest void content (1%) and fastest filling times. The interfacial and mechanical study revealed that bio-epoxy resin is highly comparable with epoxy and polyester resin systems. A noticeable difference was observed in the failure modes of composite laminates with three resin systems. Epoxy composites, for instance, formed a relatively stronger bond with the flax fibers, leading to more fiber breakage in tension. In contrast, polyester composites exhibit more fiber pull-out and debonding. Bio-epoxy composites show mixed modes of failure, if not more similar to epoxy composites. Similarly, polyester composite exhibit more ductile behavior, with relatively larger failure strains in tension and flexure, but also substantially greater dent depth in falling weight impact tests. Overall bio-epoxy composites proved to be a suitable alternative to conventional resins in terms of mechanical behavior.

## Figures and Tables

**Figure 1 materials-13-05485-f001:**
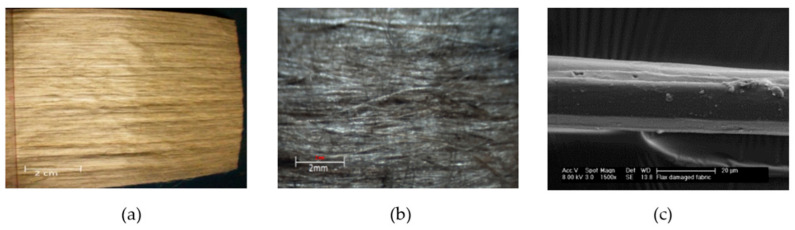
(**a**) Digital photographic image of flax tape. (**b**) Optical micrograph of flax tape (×10). (**c**) Scanning electron micrograph (×1500) of a flax fiber from the tape.

**Figure 2 materials-13-05485-f002:**
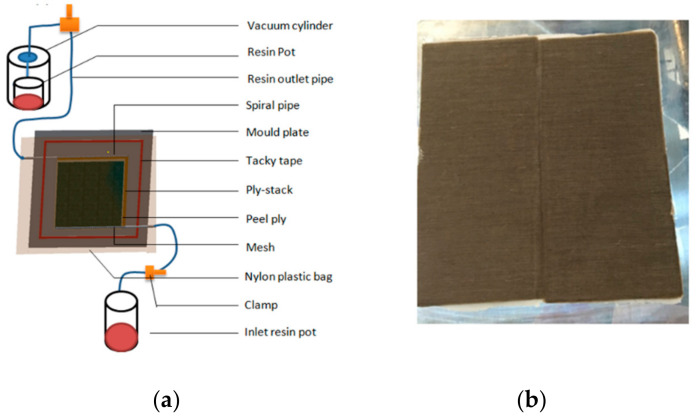
(**a**) Vacuum resin infusion process used for manufacturing composites; and (**b**) fabricated cross-laminated flax fiber tape composites.

**Figure 3 materials-13-05485-f003:**
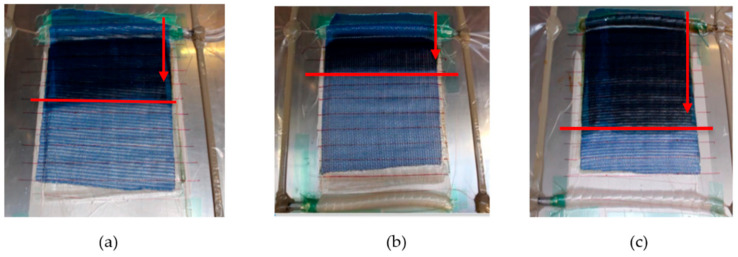
Experimental photograph of resin flow during (**a**) B-flax, (**b**) E-flax and (**c**) P-flax composite manufacturing. The red line marks the flow front position at two minutes fill time.

**Figure 4 materials-13-05485-f004:**
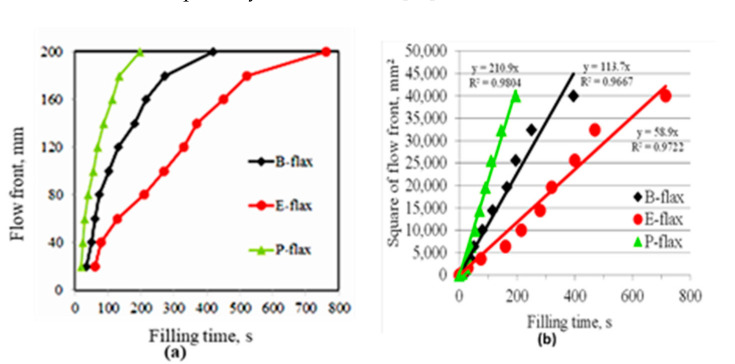
(**a**) Flow front (mm) versus filling time (s) and (**b**) square of flow front (mm^2^) versus filling time (s) of cross-laminated flax tape impregnated with three different resin systems. The slope of the graph *x*^2^/*t* (right) can be used as an input in Equation (1) to determine *K_unsat_* for the flax tape reinforcements.

**Figure 5 materials-13-05485-f005:**
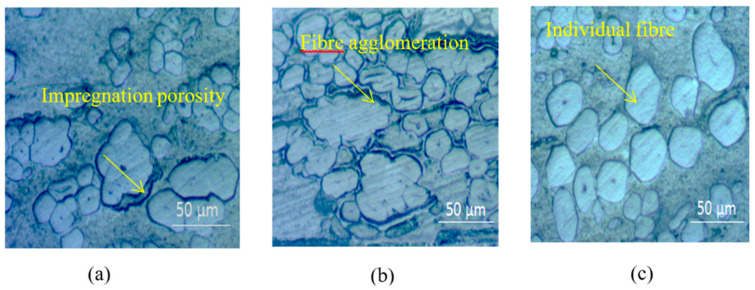
Optical micrographs for cross-sections of (**a**) B-flax, (**b**) E-flax, and (**c**) P-flax composites (×500).

**Figure 6 materials-13-05485-f006:**
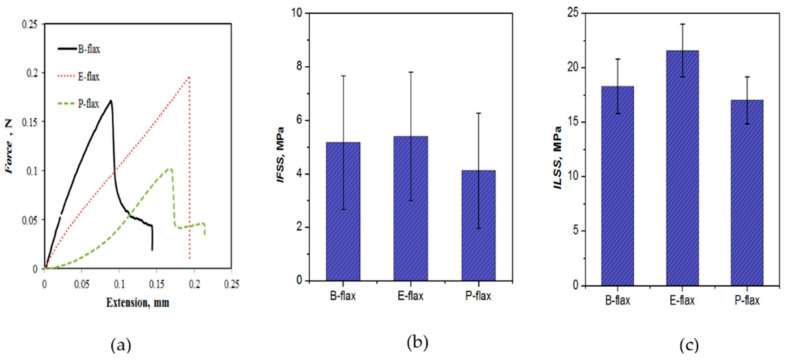
(**a**) Typical force-extension curve in the micro-bond test. (**b**) Interfacial shear strength (IFSS) of flax fiber resin interfaces measured through micro-bond testing. (**c**) Interlaminar shear strength (ILSS) of flax fiber composites with different resin systems.

**Figure 7 materials-13-05485-f007:**
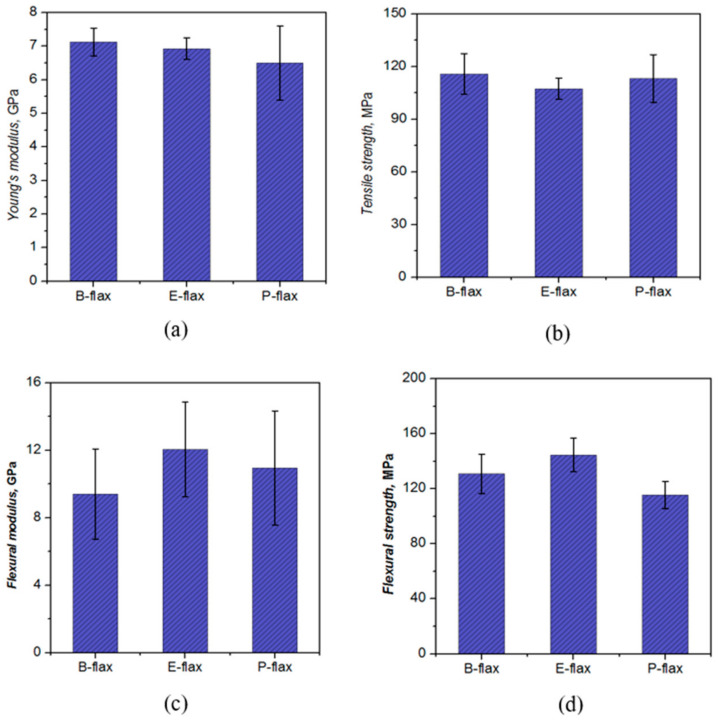
Mechanical properties of the composites using three resin systems (**a**) Young’s modulus, (**b**) tensile strength, (**c**) flexural modulus and (**d**) flexural strength.

**Figure 8 materials-13-05485-f008:**
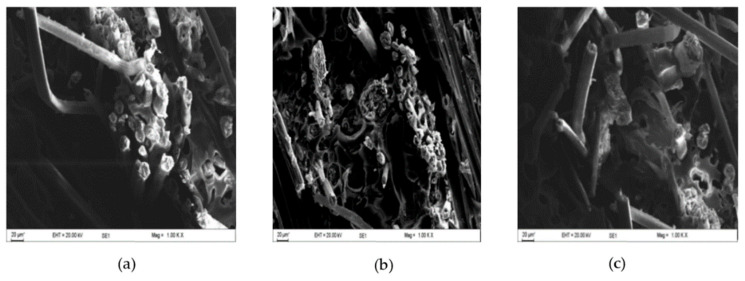
SEM micrograph of fractured specimen of flax fibre composites (**a**) B-flax, (**b**) E-flax and (**c**) P-flax, at 250× magnification.

**Figure 9 materials-13-05485-f009:**
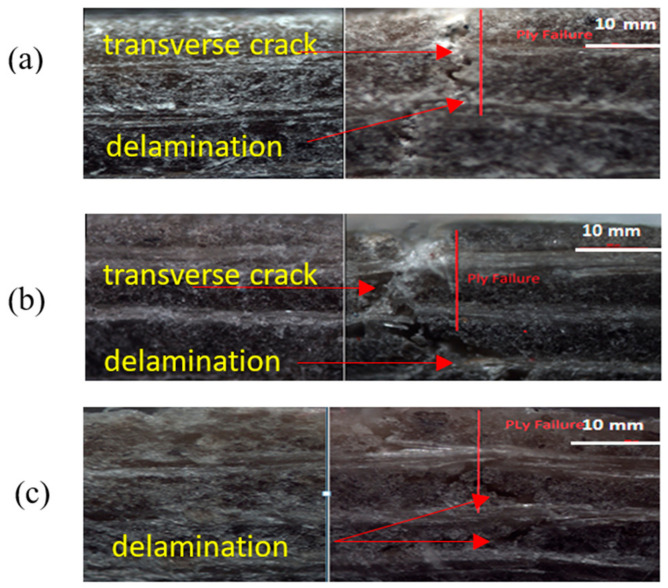
Optical micrograph of fractured specimen of flax fibre composites (**a**) B-flax, (**b**) E-flax and (**c**) P-flax at 50× magnification.

**Figure 10 materials-13-05485-f010:**
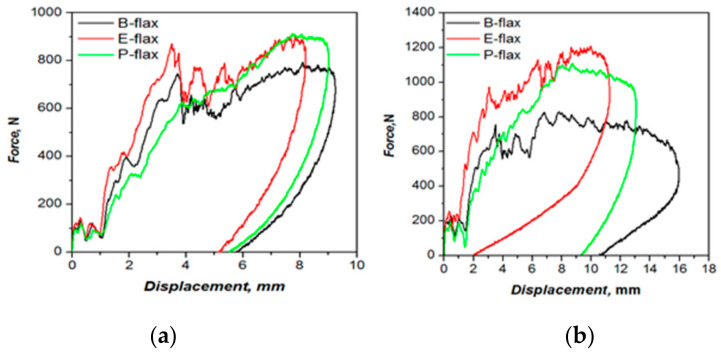
Force versus displacement curve of flax fibre composites using three resin system (**a**) 5 J, (**b**) 10 J.

**Figure 11 materials-13-05485-f011:**
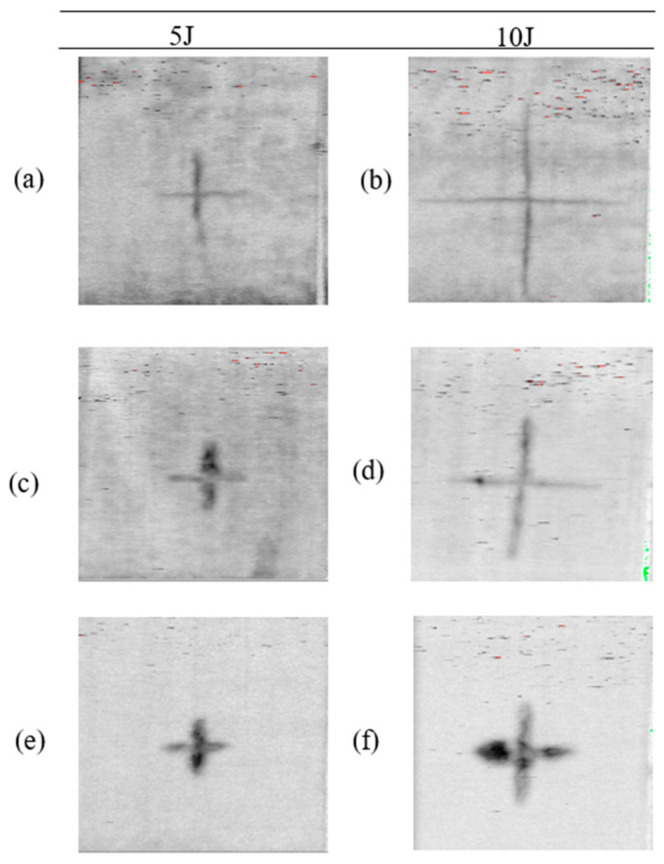
Damage area from CT-scanned image in composites (**a**) B-flax at 5 J, (**b**) B-flax at 10 J, (**c**) E-flax at 5 J, (**d**) E-flax at 10 J, (**e**) P-flax at 5J and (**f**) P-flax at 10 J.

**Figure 12 materials-13-05485-f012:**
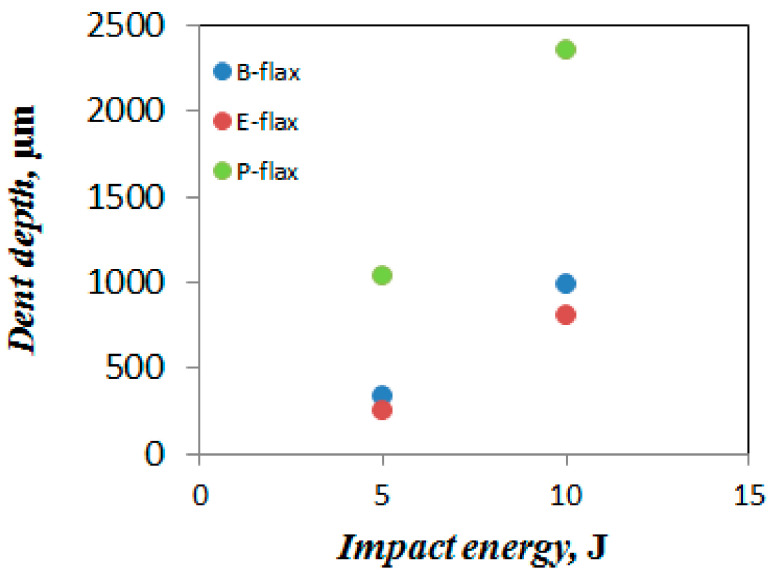
Comparison of the dent depth of flax tape thermoset composites after impact test with 5 and 10 J.

**Table 1 materials-13-05485-t001:** Resin properties (from supplier datasheets).

Resin	Viscosity (mPas)	Gel Time at 25 °C (min)	Cured Density *ρ_m_* (gcm^−3^)	Tensile Modulus *E_m_* (GPa)	Tensile Strength *σ_m_* (MPa)	Failure Strain *ϵ**_m_* (%)
Bio-epoxy	800	90	1.1–1.2	3.0	68	6
Epoxy	1200	95	1.15	3.15	70	6
Polyester	160	85	1.12	3.58	66	2.5

**Table 2 materials-13-05485-t002:** Physical properties of flax fiber tape composites.

Reinforcement Type	Resins	Fibre Weight Fraction *w_f_* (%)	Composite Density *ρ_c_* (%)	Fibre Volume Fraction *v_f_* (%)	Void Volume Fraction *v_p_ (*%)
B-flax [0.90]_12_	Bio-epoxy	36	1.22	30	~2
E-flax [0.90]_12_	Epoxy	34	1.21	27	~2
P-flax [0.90]_12_	Polyester	37	1.25	30	~1

**Table 3 materials-13-05485-t003:** Tensile properties of flax fiber composites (mean ± stdev).

Reinforcement Type	Resin	Fibre Volume Fraction *v_f_* (%)	Composite Tensile Modulus *E_c_* (GPa)	Composite Tensile Strength *σ_c_* (MPa)	Composite Failure Strain *ε_c_* (%)
B-flax	Bio-epoxy	30	7.12 (± 0.42)	115.8 (± 11.5)	1.93 (± 0.12)
E-flax	Epoxy	27	6.92 (± 0.32)	107.4 (± 6.0)	1.85 (± 0.05)
P-flax	Polyester	30	6.50 (± 1.10)	113.2 (± 13.7)	2.02 (± 0.06)

**Table 4 materials-13-05485-t004:** Flexural properties of flax tape composites.

Sample Co	Flexural Strength (MPa)	Flexural Modulus (GPa)	Flexural Strain (%)
B-Flax	130.9 (±14.3)	9.4 (±3.7)	2.68 (±0.17)
E-Flax	144.5 (±12.2)	12.1 (±1.4)	2.81 (±0.03)
P-Flax	115.6 (±9.9)	11.0 (±0.7)	3.38 (±0.07)

**Table 5 materials-13-05485-t005:** Parameters obtained from impact test on different composites.

Specimen	Peak Force (N)	Maximum Displacement (mm)	Absorbed Energy (J)
**Energy: 5 J**			
B-flax	78,449 ± 71.86	5.89 ± 1.89	4.31 ± 0.33
E-flax	91,471 ± 54.82	5.48 ± 0.79	4.23 ± 0.18
P-flax	91,874 ± 48.38	5.28 ± 1.03	4.36 ± 0.15
**Energy: 10 J**			
B-flax	85,016 ± 6.40	10.53 ± 2.82	8.49 ± 0.80
E-flax	124,609 ± 42.42	4.15 ± 1.30	8.03 ± 0.015
P-flax	110,462 ± 69.73	9.08 ± 0.30	9.30 ± 0.45

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
