# Peer review of "Mechanical Properties of Flax Tape-Reinforced Thermoset Composites"

_materials, 2020, doi:10.3390/ma13235485_

Round 1
Reviewer 1 Report
The subject of this work is appropriate for publication in Materials. Some minor revisions described below are recommended prior to publication.
(1) line 174 - 175, Figure 3: The flow direction of resin should be clearly shown in the figure.
(2) line 259 - 261, Figure 6(a): The further detailed explanation on the figure (appearance of the dynamic friction) should be added to help understanding.
(3) Spelling, description of punctuations, proper use of superscript/subscript, and usage of capital/lowercase letters should be carefully re-checked and corrected.
(4) References: Format of the citation should be adjusted to that of this journal. The abbreviated and fully spelled journal names are mixed. Information on the ref. (30) seems to be partly lacked (....;).
Reviewer 2 Report
The paper entitled “Mechanical Properties of Flax Tape Reinforced Thermoset Composites” is an interesting article.
In this research article the authors proposed the development and the characterization of three thermoset resin systems - bio-epoxy, epoxy, and polyester - with 30 v% flax fibre reinforcement have been studied to identify the optimal fibre-resin combination in a typical composite structure. The authors proposed several characterizations astensile, interface and interlaminar shear strength together with flexural and impact damage tolerance were compared in this study.
The data highlighted that mechanical and interfacial properties were not significantly affected by the different resin systems. At the end overall, bio-epoxy resin demonstrates comparable performance to epoxy and polyester resins for use in (semi-)structural bio-composites.
The paper is interesting, a lot of characterizations are presented in this work. I have no hesitation to suggest the publication of this manuscript after minor revision. The manuscript will be published in Materials Journal.
Specific comments
Figure: The authors are invite to maintain the same dimensions for all the graphs and images reported in the same Figure.
Flexural mechanical properties-paragraph 3.7: the authors are invited to explain better the effect of flax into different matrices in terms of mechanical performance.
Reviewer 3 Report
- Line 341: “Flexural properties obtained in three-point bending test from various flax tape composites are listed in Table 4 and Figure78c,d.”,Please check “Figure78”.
- On Figure 9,please mark "reverse cracking and delamination damage”.
- “the results in Figure 9(a,b) shows that transverse cracking and delamination damage occurred more readily in the laminate made from bio-epoxy and epoxy composites than the polyester composites”, but the Flexural strength of the laminate made from bio-epoxy and epoxy composites is higher than that of the polyester composites(Table 4), Is it contradictory? Please explain the damage mechanism in detail.
- Generally speaking, interface strength and interface toughness are contradictory properties. High interface performance can withstand higher crack initiation load, while high interface toughness has better damage tolerance. If possible, it is suggested to compare the interface strength and interfacial toughness of three different matrix composites.
- Fiber reinforced composites are materials with different tensile and compressive properties. Why not do uniaxial compression tests?
